# Role of Dipeptidyl Peptidase 4 Inhibitors in Antidiabetic Treatment

**DOI:** 10.3390/molecules27103055

**Published:** 2022-05-10

**Authors:** Ruili Yin, Yongsong Xu, Xin Wang, Longyan Yang, Dong Zhao

**Affiliations:** Beijing Key Laboratory of Diabetes Prevention and Research, Center for Endocrine Metabolic and Immune Diseases, Beijing Luhe Hospital, Capital Medical University, Beijing 101149, China; 13683094519@163.com (R.Y.); xuyongsong@ccmu.edu.cn (Y.X.); wangxin6305@163.com (X.W.); lyyang15@ccmu.edu.cn (L.Y.)

**Keywords:** DPP4i, T2DM, GLP1, GIP

## Abstract

In recent years, important changes have occurred in the field of diabetes treatment. The focus of the treatment of diabetic patients has shifted from the control of blood glucose itself to the overall management of risk factors, while adjusting blood glucose goals according to individualization. In addition, regulators need to approve new antidiabetic drugs which have been tested for cardiovascular safety. Thus, the newest class of drugs has been shown to reduce major adverse cardiovascular events, including sodium-glucose transporter 2 (SGLT2) and some glucagon like peptide 1 receptor (GLP1) analog. As such, they have a prominent place in the hyperglycemia treatment algorithms. In recent years, the role of DPP4 inhibitors (DPP4i) has been modified. DPP4i have a favorable safety profile and anti-inflammatory profile, do not cause hypoglycemia or weight gain, and do not require dose escalation. In addition, it can also be applied to some types of chronic kidney disease patients and elderly patients with diabetes. Overall, DPP4i, as a class of safe oral hypoglycemic agents, have a role in the management of diabetic patients, and there is extensive experience in their use.

## 1. Introduction

Dipeptidyl peptidase 4 (DPP4) enzyme is a type II transmembrane glycoprotein, expressed ubiquitously in many tissues, including the immune cells, kidney, liver, pancreas, fat cells, and presents as a soluble form in the circulation [1]. Dipeptidyl peptidase 4 is a serine protease, can cleave and inactivate incretin hormones, glucagon-like peptide 1 (GLP-1), glucose-dependent insulinotropic polypeptide (GIP), neuropeptides, and chemokines [2]. In addition, DPP4 has been shown to have a direct pro-inflammatory role in lymphocytes, macrophages, and smooth muscle cells [3,4]. Dipeptidyl peptidase 4 plays a major role in glucose and insulin metabolism, but its functions are not fully understood yet. On one hand, DPP4 degrades incretins such as GLP-1 and GIP, ultimately leading to reduced insulin secretion and abnormal visceral adipose tissue metabolism; on the other hand, DPP4 regulates postprandial glucose through degradation of GLP-1. Due to its ability to prevent the inactivation of GLP-1, DPP4 inhibition (DPP4i) was explored as a target for the treatment and management of type 2 diabetes mellitus (T2DM) in the 1990s [5,6,7].

T2DM is the most common type of diabetes and associated with a low-grade chronic inflammation induced by the excessive visceral adipose tissue. This inflammatory status results in dysregulation of homeostatic glucose regulation and peripheral insulin sensitivity. Dipeptidyl peptidase 4 activity is correlated with the onset and severity of obesity and diabetes [8]. The levels of plasma DPP4 activity are elevated in diseases, including T2DM [9], obesity [9], chronic diabetic kidney disease [10], cardiovascular diseases [7] and atherosclerosis [11]. Recently, the circulating levels of endogenous soluble DPP4 were found to be dissociated from the extent of systemic inflammation, glucose intolerance and white adipose tissue inflammation [12]; therefore, the search for DPP4 inhibitors is a viable approach. Sitagliptin may have potential therapeutic agent for the treatment of cardiovascular diseases via suppressing activation of p38/NF-κB signaling [7].

This review discusses the role of DPP4i in the treatment of diabetes, highlighting their benefits and risks. The article will focus primarily effect of the approved DPP4i: sitagliptin, vildagliptin, saxagliptin, alogliptin, and linagliptin.

## 2. DPP4 and DPP4 Inhibitions in Diabetes

### 2.1. Mechanisms of Effect of DPP4i

Diabetes mellitus (DM) is a worldwide health problem, which is a major cause of blindness, chronic kidney disease (CKD), stroke, lower extremity amputations, coronary heart disease and heart failure (HF) [13]. T2DM has changed from a chronic disease of the elderly in the traditional concept to a chronic disease of middle-aged and even children and adolescents [14,15]. Excess body fat along with age constitute the two most important risk factors for the premature development of T2DM [15,16]. Early onset T2DM relative to late-onset disease is associated with a more rapid deterioration of β-cell function, emphasizing the importance for early diagnosis and treatment initiation [17]. Obesity-related mechanisms that are potentially linked to the severity of the disease include adipocyte lipid spillover, ectopic fat accumulation and tissue inflammation [18]. Therapies aiming to decrease body weight are consequently a valuable strategy to delay the onset and decrease the risk of T2DM, as well as managing established disease [19]. 

In the past few decades, drug therapy for T2DM has developed greatly and involves several new strategies [20,21,22]. These new strategies include more patient-friendly ways to use the drug, such as improving weight loss. However, animal studies have demonstrated that a key barrier to the development of anti-obesity drugs is the large inability to predict human cardiovascular safety [23,24,25]. In tolerable doses, they rarely achieve 10% weight loss. Although the clinical success of these agents has laid the foundation for a new era of anti-obesity drugs, there is considerable debate as to how GLP1/GIP regulates metabolism and whether its receptor agonists or antagonists can be the drugs of choice for treating obesity and T2DM. At present, DPP4 inhibitors are widely used for the treatment of T2DM [26,27,28,29]. The basis for this approach lies with the finding that DPP4 has a key role in determining the clearance of the incretin hormone, GLP1 [5]. GLP1 is an intestinal peptide, which was known to have a role in glucose homeostasis via actions that include the potentiation of glucose-induced insulin secretion and the suppression of glucagon secretion [30]. 

Dipeptidyl peptidase 4 inhibitor (DPP4i) itself has no hypoglycemic activity. Instead, their anti-hyperglycemia effect is achieved primarily by altering levels of endogenous substrates. Once the catalytic activity of DPP4 is inhibited, the levels of these substrates change. To date, GLP1 has been considered to play a major role in the therapeutic effect of DPP4i [23]. GLP1 has been shown to be a physiological DPP4 substrate [23,25]. In vivo, endogenous levels of intact, biologically active peptides increase with DPP4 inhibition and are associated with improved glucose homeostasis [31,32]. Some studies found that GLP1 receptor antagonist inhibited GLP1 signaling pathway, and the hypoglycemic effect of DPP4i decreased [33,34], thus confirming the role of GLP1 in the mechanism of action of DPP4i. It also indicates that GLP1 is not the only regulatory factor, and even in the absence of GLP1 receptor activation, the hypoglycemic activity of DPP4i is still significant [33,34].

Another physiological substrate of DPP4 is glucose-dependent insulin polypeptide (GIP), also known as incretin, and the level of GIP increases with inhibition of DPP4 activity [35,36]. Similar to GLP-1, GIP enhances insulin secretion in pancreatic beta cells in a glucose-dependent manner but appears to act in a different way on glucagon secretion [37,38]. The response to GIP was also impaired in T2DM patients. In the past, views on the possible role of GIP in the treatment of T2DM have been largely ignored, because early studies have shown that GIP′s ability to stimulate insulin secretion is severely impaired. However, in T2DM patients, further studies to explore this problem were unable to be carried out due to the lack of appropriate GIP receptor antagonists. Recent studies have shown that GIP can improve glycemic control in patients with T2DM [39,40] and have revived studies on the development of novel antagonists [41,42]. These studies have led to a re-evaluation of the role of GIP in the anti-hyperglycemia of DPP4i. In addition, GLP1′s ability to inhibit glucagon secretion is weakened when blood glucose levels drop below normal fasting levels, while GIP enhances glucagon response to hypoglycemic levels. Thus, during insulin-induced hypoglycemia, glucagon secretion is increased due to GIP use [43]. Therefore, the increase in intact GIP levels observed after inhibition of DPP4 may help maintain the counter-regulatory response of glucagon when glucose levels are controlled at hypoglycemia [44,45]. Thus, GIP′s role in improving glucagon counter-regulation may further contribute to reducing the risk of hypoglycemia associated with DPP4i.

Recent studies found the direct or indirect role of soluble DPP4 in brain, gastric, liver, kidney, adipose tissue, pancreas (with islet), cardiovascular system and muscle through GLP1/GIP signaling (Figure 1). However, whether other DPP4 substrates also contribute to the therapeutic effect of DPP4i remains to be determined. In vitro, many peptide hormones and chemokines are susceptible to DPP4 cleavage when incubated with DPP4 at high concentrations [25,45]. However, there is not much evidence that they are altered in vivo by DPP4i and there have been no adverse reactions or safety issues caused by off-target effects of DPP4i on other endogenous substrates [46,47].

### 2.2. DPP4 Inhibitors

When DPP4 was identified as a therapeutic target, the search began for compounds suitable for clinical use, namely the progressive development of DPP4 inhibitors such as sitagliptin [48] and saxagliptin [49]. Currently, several structures oriented to target-specific interaction with DPP-4 are already known and officially approved by the United States Food & Drug Administration (FDA), including sitagliptin [50], saxagliptin [51], alogliptin [52], and linagliptin [53], and vildagliptin 12801240 is authorized in Europe (Table 1). 

**Sitagliptin.** Sitagliptin was the first DPP4i to receive marketing approval. The apparent terminal elimination half-life of sitagliptin is 12.4 h and renal clearance is 350 mL/min [26,54]. In healthy adult volunteers, sitagliptin is rapidly absorbed orally after a single 100 mg dose and reaches peak plasma concentration 14 h after the dose. The pharmacokinetic characteristics of sitagliptin in T2DM are generally similar with those of healthy volunteers [26,54]. Earlier results showed that sitagliptin was mainly eliminated by renal excretion, and the renal clearance rate accounted for about 70% of the plasma clearance rate of sitagliptin in healthy volunteers [55]. Absolute bioavailability of sitagliptin is 87%, oral absorption is not affected by food, and drugs can be taken regardless of food [26,54,56].

**Vidagliptin.** Vildagliptin was the second DPP4i, which to be approved by Europe [57]. About 57% of the circulation of Vildagliptin in vivo is cytochromatin-independent, with a large amount of hydrolysis to produce an inactive molecule (LAY151). The remaining 18% is circulated as an active drug [58,59]. Therefore, compared with sitagliptin, it has a shorter half-life (~2 h) and is administered in a twice-daily regimen. This metabolism is the main route of elimination of maternal drugs; however, LAY151 is cleared by the kidney and administered in a manner that increases the risk of exposure in patients with impaired renal function [60].

**Saxagliptin.** Saxagliptin is an effective anti-diabetes drug, which expands the inhibitory effect of DPP4 enzyme [61] and metabolized via cytochrome P450 3A4/A5 [62]. The use of saxagliptin at a dose of 2.5 mg, poor membrane permeability and solubility in water may further lead to its elimination with a short half-life (4–6 h) and therefore need to be dosed more than once daily [25]. The parent molecules of saxagliptin are cleared by metabolism in the liver and its metabolites are cleared by the kidneys [61,63]. However, the effect of liver damage on drug exposure is small, meaning that the therapeutic dose does not need to be changed; however, consistent with the elimination of DPP4i in other kidneys, some dose reductions are recommended when renal function declines [64].

**Linagliptin.** Linagliptin is the latest to come to market, approved for glycemic management of type 2 diabetes [65,66,67]. The metabolism of Linagliptin is not obvious, and its half-life is long (effective half-life is about 12 h, terminal decay is greater than or equal to 100 h), but compared with other DPP4i, the kidney plays a very small role in its elimination, only less than 6% of the drug is cleared in the kidney, most of the drug is excreted into bile then eliminated in the feces [68,69,70]. Therefore, linagliptin is not affected by changes in renal function and dose is not adjusted according to renal function [65]. Although it is eliminated by the biliary pathway, there is no clinically significant change in liver damage to drug exposure nor dose adjustment [71]. In the latest randomized trial, in patients with T2D and high cardiovascular (CV) risk, linagliptin showed non-inferiority compared with placebo in the risk of major CV events, with a median time of 2.2 years [72]. Linagliptin is well tolerated in Asian T2DM, with low risk for adverse events [73].

**Alogliptin.** Like sitagliptin, it has no significant metabolism and has a half-life of about 12.4–21.4 h [52,74]. Due to a long half-life, alogliptin is generally prescribed once a day. Alogliptin is cleared primarily by the kidney through glomerular filtration and active secretion mechanisms; therefore, it is recommended to reduce the dose in patients with reduced renal function [74]. Alogliptin alone or in combination with metformin, pioglitazone, glyburide, or insulin significantly improved glycemic control in adults or elderly T2DM compared with placebo. Alogliptin will primarily be used to avoid hypoglycemic events in patients with congestive heart failure, kidney failure and liver disease, as well as in the elderly. Alogliptin protects against cyclophosphamide induced lung toxicity by reducing oxidation, inflammation and fibrosis, making it a promising pharmacological treatment for reducing lung toxicity [75].

### 2.3. Benefits of DPP4i

As above discussed, the efficacy of DPP4i in inhibiting the catalytic activity of DPP4 is clearly related to its efficacy as an antidiabetic agent. Marketed in the United States, DPP4i has also been evaluated for cardiovascular safety in large cardiovascular outcome trials (CVOTs) and neither type of DPP4i increases the risk of major adverse cardiovascular events [72,76]. DPP4i reduces long-term cardiovascular risk after percutaneous coronary intervention in patients with diabetes via the insulin-like growth factor-1 axis [77]. However, in CVOT′S there was not any cardiovascular benefit and saxagliptin increase heart failure hospitalization, vildagliptin is not marketed in the USA, so there is no data associated with CVOT (Table 2). 

Previously discussed reports suggest that DPP4i may be more effective in the Asian population than in the white population [78,79]. It has been suggested that this difference may be related to pathological differences in T2DM between the two groups (emaciation and impaired beta cell function phenotype in Asian patients and obesity and insulin resistance phenotype in white patients) [80,81]. DPP4i all benefit by being highly orally available and well tolerated anti-hyperglycemic medications. They are also easy to use, requiring no dose titration and can be taken at any time of day without regard to mealtimes. Sitagliptin, alogliptin, and linagliptin interact noncovalently with residues in the catalytic bag, then decompose unchanged as parent inhibitor molecules, and then interact freely with the enzyme again plus their inherent long half-life, resulting in sustained DPP4 inhibition, which is compatible with a once-daily dosing regimen [82,83]. While saxagliptin is covalently bound by cyanopyrrolidine fragments, prolongating the inhibitor′s interaction with the enzyme until hydrolysis releases the major metabolite 5-hydroxysaxagliptin [82,83]. Accordingly, in short-term studies (1–9 days in duration), head-to-head comparisons in patients with T2DM have shown that when used at their therapeutic doses, sitagliptin (100 mg once daily) and saxagliptin (5 mg once daily) all achieve the same maximal and trough levels of DPP4 inhibition [84] and are associated with similar enhancements of intact incretin hormone concentrations [85,86]. It follows, therefore, that if the extent and duration of DPP4 inhibition is similar, the improvement in glycemic control should also be similar. Indeed, in several studies, direct comparisons have been made between DPP4 inhibitions, glucose excursions [85,86] and HbA1c levels [85,87] are reduced to similar extents (Table 3). These inhibitors achieve similar degrees of DPP4 inhibition to the shorter-acting DPP4i discussed already, in line with this finding, head-to-head comparisons have also shown them to be non-inferior with respect to HbA1c control [88].

Their mechanism of action, involving both insulinotropic and glucagon-promoting effects, means that they combine well with other anti-diabetic agents to give additional HbA1c-lowering efficacy. The potency of DPP4i in A1C reduction is moderate. In this regard, the individual DPP4i have a low propensity for drug–drug interactions, meaning that they can be used with any other medications without the need for dose adjustment. Similarly, doses of other agents used together with DPP4i do not generally require adjustment; however, reduction in concomitant sulfonylurea or insulin doses is recommended to minimize the hypoglycemic risk associated with sulfonylureas and insulin [58,89,90]. The discovery that metformin also stimulates GLP1 secretion, further explaining the special efficacy of metformin in combination with DPP4i [91]. In a retrospective study in Korea, metformin and DPP4i was found to be effective in reducing HbA1c below 7%, with a low incidence of hypoglycemia [92].In tacrolimus-induced SD rat model and nephrotoxicity test, DPP4i and sodium–glucose transporter 2 inhibitors (SGLT2i) reduced blood glucose level and HbA1C level, and increased plasma insulin level and islet size of rats, and improved renal function and reduced interstitial fibrosis and pro-fibrotic cytokines, which providing a theoretical basis for the combination of SGLG2i and DPP4i in the treatment of tacrolimus-induced DM and nephrotoxicity [93]. The dual effect of DPP4i on α- and β- cells means that they also bind well to the islet independence of SGLT2i [94]. In addition, when DPP4i is combined with insulin secretin or insulin itself, DPP4i provides a complementary effect due to its inhibition of glucagon secretion and reduction in hepatic glucose production, which also means that a beneficial effect on glucose control can be achieved even with reduced β cell function [94]. In Chinese trials with T2DM, the addition of linagliptin and insulin improved glycemic control and was well tolerated with no increased risk of hypoglycemia or weight gain [95].

As described earlier, the effect of DPP4i on the internal environment of glucose is not direct, mediated through the action of the substrates they protect, especially GLP1. Therefore, considering that the activity of DPP4 is already completely inhibited when DPP4i is used at its therapeutic dose, any increase in exposure to the drug will not have a further hypoglycemic effect (since the enzyme cannot be inhibited by more than 100%). Combined with the fact that the role of GLP1 and GIP is itself glucose dependent, the inherent risk of treating hypoglycemia with DPP4i is particularly low [96,97]. Therefore, DPP4i is particularly suitable for use in elderly, frail and/or vulnerable patients whose long-term type 2 diabetes and its complications often result in multidrug therapy. In addition, patients with liver and kidney damage may be contraindications to other antidiabetic drugs [98]. Studies have shown that DPP4i is safe and effective in elderly patients with T2DM (65 or 70 years and above), and CVOTs including elderly patients with comorbidities (≥75 years), indicating that DPP4i is safe and effective in this population and showed a similar glycemic effect as the younger participants [99,100]. In addition, DPP4i has been shown to be effective and well tolerated in patients with renal dysfunction (including end-stage renal disease patients on dialysis) and DKD [101,102]. Since sitagliptin, alogliptin, and saxagliptin are cleared through the kidney, it is recommended that DPP4i doses be reduced [64,74,103] in patients with reduced renal function, while linagliptin is cleared through the biliary tract without dose adjustment [65]. Dose adjustment for renal elimination of DPP4i was based on pharmacokinetics rather than safety concerns. Therefore, any increase in dose will not cause hypoglycemia or other mechanism-based adverse reactions because the enzyme is already maximally inhibited, and compound specific adverse reactions are unlikely; earlier dosing studies found no adverse events [55,104,105] when using 4–32 times the therapeutic dose. 

### 2.4. Anti-Inflammation Effects of DPP4i

DPP4 plays an important role in the maturation and activation of T cells and immune responses, with independent catalytic activity [106,107]. DPP4, released from hepatocytes and adipose tissue and exogenously administered, promotes inflammatory responses in multiple tissues, often associated with the development of insulin resistance [12,107]. In vivo sitagliptin can protect the endothelial function of renal artery in spontaneously hypertensive rats by GLP-1 signaling [103]. DPP4 inhibitors play a direct role in anti-atherosclerosis by improving endothelial dysfunction, inhibiting inflammation and oxidative stress, and improving plaque instability [104,105]. Our previous studies have demonstrated that sitagliptin may protect diabetic fatty liver by reducing ROS production and NF-κB signaling pathway activation [106]. Recent studies found the anti-inflammation of DPP4 inhibitors in non-diabetes and diabetes model, which summaries in Table 4. Although DPP4i has shown many anti-inflammatory effects in diabetes complications and other disease, this effect has not been shown in clinical trials. Therefore, more human population studies are needed to verify the anti-inflammatory effects of DPP4 in liver, lung, heart, kidney, and nerve in the future.

### 2.5. Adverse Effects

FAERS data mining helped examine adverse events associated with DPP-4 inhibitors, all of which were disproportionately associated with four types of adverse events: “gastrointestinal nonspecific inflammation and dysfunction”, “allergy”, “severe skin adverse reactions”, and “noninfectious diarrhea” [148]. As for the analysis of the level of preferred terms specified, DPP4i was associated with higher reports of gastrointestinal, pancreatic, malignancies, infections, musculoskeletal disorders, systemic diseases, allergies, and cutaneous adverse reactions [149].

## 3. Conclusions

Dipeptidyl peptidase 4 plays a key role in the regulation of glucose metabolism. The DPP4 inhibitors are effective and safe hypoglycemic therapy for type 2 diabetes that are effective orally and associated with a low risk of hypoglycemia, weight gain, or other adverse events on a solid basis. There is also a large body of clinical and experimental data suggesting that improved islet function is the key mechanism behind the inhibitory hypoglycemic effect of DPP4i, both of which are associated with increased insulin secretion and glucagon secretion inhibition. 

Additionally, DPP4i′s ability to reduce the risk of hypoglycemia and to work in complementarity with other antidiabetic drugs makes them widely used second-line drugs and means they are particularly useful for other drugs or contraindications that may not be preferred. Co-action of DPP4i with other drugs such as metformin, SGLT2i, and pioglitazone can provide additional glycemic efficacy without increasing the burden of pills [150,151,152]. However, unlike GLP1 receptor agonists and SGLT2i, DPP4i was insignificant in terms of cardiovascular benefit and reduced risk of major adverse cardiovascular events. In summary, DPP4i is safe and effective in the majority of T2DM patients, and we hope that DPP4i can help patients achieve glycemic goals in an overall favorable therapeutic setting. In addition, with the deepening of some DPP4i related research, more excellent efficacy may be developed.

## Figures and Tables

**Figure 1 molecules-27-03055-f001:**
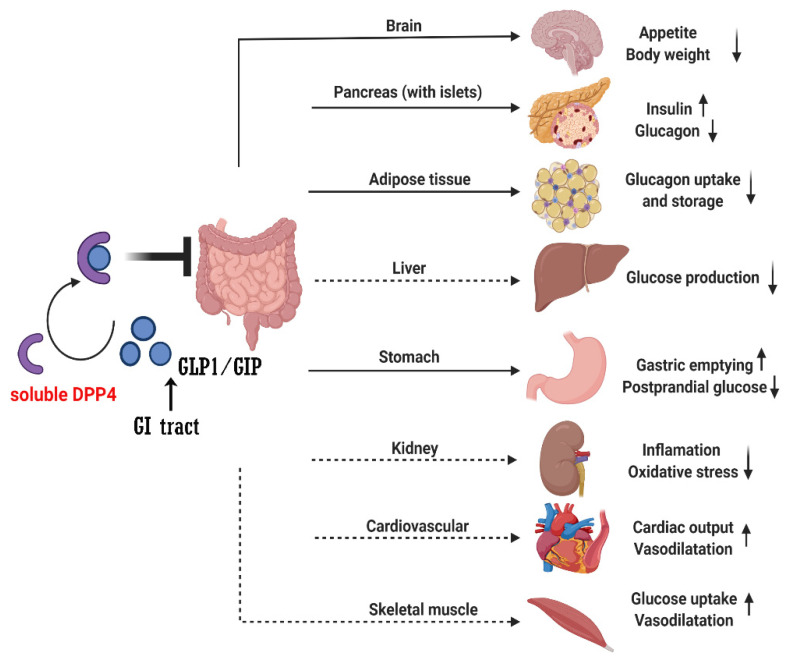
The action of DPP4 degrades the GLP1 and GIP. GI: gastrointestinal; GIP: glucose-dependent insulinotropic polypeptide, GLP1: glucagon-like peptide-1, DPP4: dipeptidyl peptidase-4. Solid arrows mean the direct regulation effects by GLP1 or GIP; dashed arrows mean the indirect regulation effects by GLP1 or GIP.

**Table 1 molecules-27-03055-t001:** The features of approved dipeptidyl peptidase 4 inhibitors (DPP4i).

DPP4i	Chemistry	Metabolism	Half-Life	Elimination Method
Sitagliptin	β-amino acid based	Minimal	12.5 h	Predominantly (>80%) renal
Vildagliptin	Cyanopyrrolidine	Hydrolysis (cytochromeindependent) to form an inactive metabolite	~2 h	Metabolism (parent) and renal (metabolite)
Saxagliptin	Cyanopyrrolidine	Hydrolysis (cytochrome P450 3A4 or P450 3A5) to form an activemetabolite	2.5 h (parent), 3 h (metabolite)	Metabolism (parent) and renal (metabolite)
Alogliptin	Modified pyrimidinedione	Minimal	20 h	Predominantly (>70%) renal
Linagliptin	Xanthine based	Minimal	~12 h (effective), >100 h (terminal)	Predominantly biliary (<6% renal)

**Table 2 molecules-27-03055-t002:** Cardiovascular outcome trials with DPP4 inhibitors.

DPP4i	Trial (Year)	Median Follow-Up, Years	Mean/Median Age, Years	Female (Total)	BMI, kg/m^2^ *	HbA1c, mmol/mol (%) *	Baseline Metformin, %	Baseline eGFR, mL/min/ [1.73 m]^2^ *	Prior ASCVD, %	Prior CHF, %
Sitagliptin	TECOS (2015)	3.0	65	4212	30.2	55 (7.2)	81	75	100	18
				(14,523)						
Saxagliptin	SAVOR-TIMI (2013)	2.1	65	5590	31.2	64 (8.0)	69	73	78	13
				(16,492)						
Alogliptin	EXAMINE (2013)	1.5	61	1722	28.7	64 (8.0)	NA	71	100	28
				(5380)						
Linagliptin	CARMEL (2019)	2.2	66	2582	31.4	64 (8.0)	54	55	57	27
				(6979)						

NA, not available; * These are expressed as mean values.

**Table 3 molecules-27-03055-t003:** The effect of DPP4 inhibitors in HbA1c.

DPP4i	Dose (mg/Day)	HbA1c Reduction
Sitagliptin	100	0.5–1.0
Saxagliptin	5	0.5–1.0
Alogliptin	25	0.6 (mean value)
Linagliptin	5	0.5–0.7

**Table 4 molecules-27-03055-t004:** The anti-inflammatory effects of DPP4 inhibitors.

DPP4i	Experimental Model	Mechanism of the Effects	Ref.
Sitagliptin	HFD-fed diabetic mice	Inhibited fatty liver inflammation; downregulates HMGB1/TLR4/NF-κB signaling pathway	[108,109]
	Diet-induced NAFLD	Inhibited pro-fibrotic and pro-inflammatory changes	[110]
	HFD-fed rats	Ameliorated apoptosis via alleviating ROS and ER stress	[111]
	Hepatic ischemia-reperfusion rat	Modulates oxidative, nitrative and halogenative stress and inflammatory response	[112]
	High glucose-induced human renal glomerular endothelial cells	Reversed the high glucose-induced oxidative stress, inflammation, and increased permebility via regulating KLF6	[113]
	Hypoxia-induced damages in endometrial stromal cells	Suppressed the expressions of the proinflammatory cytokines including TNF-α, IL-6 and MCP-1; mitigated the activation of the p-38 MAPK and NF-κB pathways	[114]
	Severe acute pancreatitis companied with acute lung injury	Reduced oxidative stress and excessive autophagy through the p62–Keap1–Nrf2 signaling pathway	[115]
	Depressive symptoms in T2DM	No effect	[116]
	Human rheumatoid arthritis synovial fibroblasts	Increased proinflammatory cytokine production, enhanced the risk of RA development (sitagliptin and vildagliptin)	[117]
	Chlorhexidine gluconate induced peritoneal dialysis rats	Reversed the EMT process, angiogenesis, oxidative stress, and inflammation	[118]
	Low-density lipoprotein cholesterol in diabetes (REASON) Trial	Did not affect the levels of inflammatory markers	[119]
	Total body irradiation induced hematopoietic cells injury	Inhibited NOX4-mediated oxidative stress and alleviated inflammation	[120]
	Breast cancer	Reprograms tumor microenvironment via a ROS–NRF2–HO-1–NF-kB–NLRP3 axis	[121]
	Obese mice	Inhibited adipose tissue inflammation, metabolic syndrome, and fatty liver via regulation of adiponectin and AMPK levels	[122]
Vildaliptin	Rheumatoid arthritis	Increased proinflammatory cytokine IL-1β, IL-6, and IL-13 production	[117]
	Septic rats with myocardial injury	Inhibited the activation of NF-κB by promoting Nrf2 to alleviate the inflammatory response	[123]
	Acetic acid-induced colitis in rats	Inhibited the expression of lncRNA IFNG-AS1 and miR-146a, PI3K/Akt/NFκB pathway, and activated CREB and nuclear factor erythroid 2-related factor 2 (Nrf2) signaling pathways	[124]
	Carbon tetrachloride-induced liver fibrosis	Attenuates liver fibrosis by targeting ERK1/2, p38α, and NF-κB signaling.	[125]
	Bleomycin-induced pulmonary fibrosis	Attenuated inflammation and fibrosis in bleomycin-induced pulmonary tissue via inhibiting the activity of CD26/DPP4	[126]
	HFD-fed rats with impaired renal function	Attenuated insulin resistance and renal lipid accumulation-induced lipotoxicity	[127]
Saxagliptin	Chronic unpredictable mild stress induced depression in rats	Increased the incretin hormones, GLP-1 and GIP, and the activation PI3K/AKT signaling pathway	[128]
	Breast cancer	Reprogramed tumor microenvironment via a ROS–NRF2–HO-1–NF-kB–NLRP3 axis	[121]
	H9c2 cardiomyocyte cell line	Ameliorated hypoxia-induced inflammation via upregulation of Nrf2 and HO-1	[129]
	Angiotensin II kidney injury model	Improved Angiotensin II suppressed anti-inflammatory regulatory T cell and T helper 2 lymphocyte activity	[130]
	Young and old SD rats	Improved endothelial senescence by activating AMPK/SIRT1/Nrf2 signaling pathway	[131]
Alogliptin	Cyclophosphamide-induced lung toxicity in rats	Ameliorated lung toxicity by mitigating the oxidative, inflammatory, and fibrotic impacts	[75]
	Lipopolysaccharide-induced neuroinflammation in mice	Attenuated neuroinflammation through modulation of TLR4/MYD88/NF-κB and miRNA-155/SOCS-1 signaling pathways	[132]
	Cyclophosphamide-induced nephrotoxicity Wistar rats	Attenuated nephrotoxicity through modulating MAP3K/JNK/SMAD3 signaling cascade	[133]
	Fibroblast-like synoviocytes	Inhibited IL-1β-induced inflammatory response	[134]
Linagliptin	Sepsis mouse	Suppressed expressions of IL-1β and intercellular adhesion molecule 1 via a NF-κB-dependent pathway	[135]
	Acetic acid-induced colitis rats	Activated AMPK-SIRT1-PGC-1α pathway and suppressed JAK2/STAT3 signaling pathway	[136]
	LPS induced U937 cells	Inhibited inflammation around the TLR-4-mediated pathway.	[137]
	Acute kidney injury in rats	Decreased inflammatory cytokines and ROS	[138]
	Early T2DM	Not altered plasma nitrate levels	[139]
	Experimental autoimmune myocarditis mice	Suppressed oxidative stress in EAM hearts	[81]
	Trinitrobenzene sulfonic acid-evoked colitis in rats	Curbed inflammation through the suppression of colonic IL-6, TNF-α, and upregulation of IL-10	[140]
	Anti-glomerular basement membrane antibody induced in nephritis rats	Improved resolution of glomerular injury and healing in non-diabetic renal disease	[141]
	OSI-906-induced hepatic steatosis	Improved hepatic steatosis via an insulin-signaling-independent pathway	[142]
	Diabetic injured kidney	Inhibited the CRP/CD32b/NF-kB-driven renal inflammation and fibrosis	[143]
	Oxidized LDL-induced THP-1 macrophage foam cell formation	Decreased the expression of CD36 and LOX-1 and increased the expression of the cholesterol transporter ABCG1	[144]
	HFD and streptozotocin (STZ) induced diabetic rats: liver fibrosis with T2DM	Improved insulin sensitivity and lipid profile and reduced inflammatory mediators, and collagen depositions	[145]
	Atherosclerosis and T2D mice	Improved glucose tolerance and reduced hepatic inflammation but had no effect on plaque burden or atherosclerotic inflammation	[146]
	Hyperglycemic mice with stroke	Exerted a neuroprotective effect through activation of the Akt/mTOR pathway along with anti-apoptotic and anti-inflammatory mechanisms	[147]
	Mouse bone marrow macrophages	Increased M2 macrophage polarization by inhibiting DPP-4 expression and activity	[130]

## Data Availability

This review collected from open access web-source PubMed.

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
