# Peer review of "Role of Dipeptidyl Peptidase 4 Inhibitors in Antidiabetic Treatment"

_molecules, 2022, doi:10.3390/molecules27103055_

Round 1

Author Response

Response to Reviewer 1 Comments

Point 1: Find out the research gap between the current therapeutic uses of the molecules and the treatment strategy that have been used currently.

Response 1:  Thanks for your suggestion. DPP4 plays a key role in the regulation of glucose metabolism. DPP4 inhibitors are effective and safe hypoglycemic therapy for type 2 diabetes that are effective orally and associated with a low risk of hypoglycemia, weight gain or other adverse events on a solid basis. There is also a large body of clinical and experimental data suggesting that improved islet function is the key mechanism behind the inhibitory hypoglycemic effect of DPP4i, both of which are associated with increased insulin secretion and glucagon secretion inhibition. In addition, experimental and clinical evidence suggests that DPP-4i has a role in diabetic complications due to its anti-inflammatory effects.

DPP4i's ability to reduce the risk of hypoglycemia and to work in complementarity with other antidiabetic drugs makes them widely used second-line drugs and means they are particularly useful for other drugs or contraindications that may not be preferred.

Point 2: Includes more schematic diagram to explain the mechanism of actions of each molecule.

Response 2: Thanks for your suggestion. The mechanism of action among the members of DPP4 inhibitors is shown in Figure 1, the mechanism for different members are listed in Table 4. If your request has not been met, we will make more detailed improvements in the future.

Point 3: Include detail mechanisms of these molecule to be used as antiobesity agents.

Response 3: Thanks for your suggestion. The mechanism for of these molecule to be used as antiobesity agents are listed in Table 4.

Point 4: You must have some recommendations about these molecules in overall treatment strategy in the conclusion section.

Response 4: Thanks for your suggestion. We modified our conclusion section in the revised manuscript.

Reviewer 2 Report

At present, DPP4 inhibitors are the newest and most promising 74
method for the treatment of T2DM

It is not correct SGLT2-I is the newest and SGLT2 and GLP-1 analog is the most promising.

 DPP4i reduces long-term cardiovascular risk 236
after percutaneous coronary intervention in patients with diabetes via the insulin-like 237
growth factor-1 axis.

It should be clear that in CVOT'S there was not any cardiovascular benefit and Saxagliptin increase heart failure hospitalization

It should be clear that the potency in A1C reduction is only moderate.

 DPP4 inhibitors play a direct role in anti-atherosclerosis by improv- 257
ing endothelial dysfunction, inhibiting inflammation and oxidative stress, and improving 258
plaque instability [104, 105]. Our previous studies have demonstrated that 259
sitagliptin may protect diabetic fatty liver by reducing ROS production and NFκB 260
signaling pathway activation [106]. 

there is no benefit in clinical trails in human

Author Response

Response to Reviewer 2 Comments

Point 1: At present, DPP4 inhibitors are the newest and most promising method for the treatment of T2DM.

Response 1: We changed it to “At present, DPP4 inhibitors are widely used for the treatment of T2DM”.

Point 2: It is not correct SGLT2-I is the newest and SGLT2 and GLP-1 analog is the most promising.

Response 2:We changed it to “the newest class of drugs has been shown to reduce major adverse cardiovascular events, including sodium-glucose transporter 2 (SGLT2) and some glucagon like peptide 1 receptor (GLP1) analog”.

Point 3: DPP4i reduces long-term cardiovascular risk 236

after percutaneous coronary intervention in patients with diabetes via the insulin-like 237

growth factor-1 axis. 

It should be clear that in CVOT'S there was not any cardiovascular benefit and Saxagliptin increase heart failure hospitalization 

Response 3: We added that in CVOT'S there was not any cardiovascular benefit and Saxagliptin increase heart failure hospitalization.

Point 4: It should be clear that the potency in A1C reduction is only moderate. 

Response4: We added that the potency of DPP4i in A1C reduction is moderate.

Point 5: DPP4 inhibitors play a direct role in anti-atherosclerosis by improving endothelial dysfunction, inhibiting inflammation and oxidative stress, and improving plaque instability [104, 105]. Our previous studies have demonstrated that sitagliptin may protect diabetic fatty liver by reducing ROS production and NFκB signaling pathway activation [106].

there is no benefit in clinical trails in human 

Response 5:: We added that there is no benefit in clinical trails in human.

Reviewer 3 Report

            My opinion  for the reviewing of "role of dipeptidyl peptidase 4 inhibitors in antidiabetic treatment" is based essentialy on the review entitled  of the French opinion of the « haute autorité de transparence » following the evaluation of the medical service rendered for gliptin-type drugs authorized in France. This report was published in 2021 with 202 pages and should serve as a guideline for the revision of the review by Yin et al.

   https://www.has-sante.fr/upload/docs/evamed/CT-19184_GLIPTINES_REEVAL_PIC_ Avis def_CTEVAL520.pdf

Extracts (in italique) are presented here to help for the present Yin’s review.

Re-evaluation of the medical service rendered (SMR) and the improvement of the medical service rendered (ASMR) at the request of the Commission by the French Directorate of Medical, Economic and Public Health Evaluation.

Indeed, the review makes a rather old synthesis on the treatment of diabetes at the time of the introduction of gliptins in the years 2000. In fact, according to the HAS and the 2021 opinion, the treatment of diabetics must be considered on the basis of complications and associated mortality. An excerpt from this opinion is: "Type 2 diabetes (T2D) is not a separate entity and represents a group of heterogeneous diseases. Its evolution is marked by the occurrence of microvascular complications affecting in particular the ocular system, the nervous system, the renal function; but also macrovascular with an increased risk of myocardial infarction, stroke and obliterative arteriopathy of the lower limbs, which represent the 1st cause of death in patients with T2DM. In addition to coronary artery disease, for which diabetes is a risk factor, there are other mechanisms by which diabetes promotes the development of heart failure: microangiopathy, metabolic factors, fibrosis, etc."

Then the knowledge of this opinion indicates and specifies the association in bi or tri therapies with metformin and sulfonamide. Gliptins are no longer recommended as a single first-line treatment, as was the case with the first marketing authorizations. My question is : the heterogenity of the diabetic patients is a limiting factor to the efficacy and tolerace of gliptins as antidiabetic drugs ?

Limitations is presented for the gliptins tobay by the HAS in France.

« an opinion in favor of maintaining reimbursement only in combination with other antidiabetic agents (dual therapy in combination with metformin or a hypoglycemic sulfonamide or triple therapy in combination with metformin and a hypoglycemic sulfonamide or with metformin and insulin), for the following specialties

* alogliptin (VIPIDIA and VIPDOMET), except for triple therapy in combination with metformin and a hypoglycemic sulfonamide

* linagliptin (TRAJENTA and JENTADUETO)

* saxagliptin (ONGLYZA and KOMBOGLYZE)

* sitagliptin (JANUVIA/XELEVIA and JANUMET/VELMETIA)

* and vildagliptin (GALVUS and EUCREAS).

This association is not presented and discussed in the review, in fact the reduction of glycated hemoglobin is less than 1%, this data is present but not discussed as a limited medical service or efficacy in the real life of diabetic patients.

Then the review presents 3 tables with only 4 gliptins, yet a review today must develop a majority of authorized drugs. The revision of these 3 tables in 2022 is absolutely necessary with the giptins authorized in the clinic.

The chemical structures of the gliptins must be presented in a journal for publication especially in the molecules journal.

An additional table concerning the originality of the gliptins today should focussed on the anti-inflammatory effects of gliptins, as well as diabetic complications or serious adverse side effects, to support the experimental and clinical evidence of the possible future use of gliptins in combination.

Our recommendations are based on the latest published scientific advances 2020-2022.

Author Response

Response to Reviewer 3 Comments

Point 1: My opinion for the reviewing of "role of dipeptidyl peptidase 4 inhibitors in antidiabetic treatment" is based essentialy on the review entitled  of the French opinion of the « haute autorité de transparence » following the evaluation of the medical service rendered for gliptin-type drugs authorized in France. This report was published in 2021 with 202 pages and should serve as a guideline for the revision of the review by Yin et al.

https://www.has-sante.fr/upload/docs/evamed/CT-19184_GLIPTINES_REEVAL_PIC_ Avis def_CTEVAL520.pdf

Extracts (in italique) are presented here to help for the present Yin’s review.

Re-evaluation of the medical service rendered (SMR) and the improvement of the medical service rendered (ASMR) at the request of the Commission by the French Directorate of Medical, Economic and Public Health Evaluation.

Response 5: Thanks for your suggestion. We uploaded this report and revised our manuscript according this guideline. Due to vildagliptin was authorized by Europe, there is not too muck CVOT associated clinical trials, so we did not describe it in our review.

Point 2: Indeed, the review makes a rather old synthesis on the treatment of diabetes at the time of the introduction of gliptins in the years 2000. In fact, according to the HAS and the 2021 opinion, the treatment of diabetics must be considered on the basis of complications and associated mortality. An excerpt from this opinion is: "Type 2 diabetes (T2D) is not a separate entity and represents a group of heterogeneous diseases. Its evolution is marked by the occurrence of microvascular complications affecting in particular the ocular system, the nervous system, the renal function; but also macrovascular with an increased risk of myocardial infarction, stroke and obliterative arteriopathy of the lower limbs, which represent the 1st cause of death in patients with T2DM. In addition to coronary artery disease, for which diabetes is a risk factor, there are other mechanisms by which diabetes promotes the development of heart failure: microangiopathy, metabolic factors, fibrosis, etc."

Response 2: Thanks for your suggestion. We updated the latest literature on DPP4i for 2020-2022 in Table 4. Since this review only focuses on the role of DPP4i in diabetes, there is not much discussion on the specific clinical use of DPP4i. Your suggestions are very valuable to us, and we have revised them according to your suggestions.

Point 3: Then the knowledge of this opinion indicates and specifies the association in bi or tri therapies with metformin and sulfonamide. Gliptins are no longer recommended as a single first-line treatment, as was the case with the first marketing authorizations. My question is : the heterogenity of the diabetic patients is a limiting factor to the efficacy and tolerace of gliptins as antidiabetic drugs ?

Response 3: This is an interesting question. DPP4i's ability to reduce the risk of hypoglycemia and to work in complementarity with other antidiabetic drugs makes them widely used second-line drugs and means they are particularly useful for other drugs or contraindications that may not be preferred.

Point 4: Limitations is presented for the gliptins today by the HAS in France.

« an opinion in favor of maintaining reimbursement only in combination with other antidiabetic agents (dual therapy in combination with metformin or a hypoglycemic sulfonamide or triple therapy in combination with metformin and a hypoglycemic sulfonamide or with metformin and insulin), for the following specialties

* alogliptin (VIPIDIA and VIPDOMET), except for triple therapy in combination with metformin and a hypoglycemic sulfonamide

* linagliptin (TRAJENTA and JENTADUETO)

* saxagliptin (ONGLYZA and KOMBOGLYZE)

* sitagliptin (JANUVIA/XELEVIA and JANUMET/VELMETIA)

* and vildagliptin (GALVUS and EUCREAS).

This association is not presented and discussed in the review, in fact the reduction of glycated hemoglobin is less than 1%, this data is present but not discussed as a limited medical service or efficacy in the real life of diabetic patients.

Response 4: Thanks for your suggestion. We added the association in the revised manuuscript.

Point 5: Then the review presents 3 tables with only 4 gliptins, yet a review today must develop a majority of authorized drugs. The revision of these 3 tables in 2022 is absolutely necessary with the giptins authorized in the clinic.

Response 5:Thanks for your suggestion. We selected the 4 gliptins which was authorized drugs by FDA. Others authorized drugs do not discussed in this manuscript.

Point 6: The chemical structures of the gliptins must be presented in a journal for publication especially in the molecules journal.

Response 6: Thanks for your suggestion. We added the chemical structures of the gliptins in the revised manuscript.

Point 7: An additional table concerning the originality of the gliptins today should focussed on the anti-inflammatory effects of gliptins, as well as diabetic complications or serious adverse side effects, to support the experimental and clinical evidence of the possible future use of gliptins in combination.

Response 7:Thanks for your suggestion. We added the table 4 which focussed on the anti-inflammatory, diabetic complications or serious adverse side effects of gliptins.

Point 8: Our recommendations are based on the latest published scientific advances 2020-2022.

Response 8:Thanks for your suggestion. We added the latest published scientific advances 2020-2022 in the revised manuscript.

Round 2

Reviewer 1 Report

The revised version seems pretty good and I would recommend to accept the manuscript for publication with the following minor suggestion:

In the section 2.4 the title would be "Anti-inflammatory effects of DPP4i" in exchange of "Anti-inflammation effects of DPP4i"

Reviewer 3 Report

The manuscript was greatly improved by adding interesting table on inflammation in experimental models.

Please check the following lines.

L13 HBA1C/A1C

L57 first citation of abbreviations L57 and L90 DPP4i  inhibitors

L141 diseases or disease (reviewer's comment: what diseases ? please  clarify.

L334 sDDP4 abbreviation s is soluble please check the first citation of this.

conclusion "safe and effective" are cited two fold, please focus your conclusion.